# Organic Molecule Assisted Growth of Perovskite Films Consisting of Square Grains by Surface-Confined Process

**DOI:** 10.3390/nano11020473

**Published:** 2021-02-12

**Authors:** Shao Xin Yan, Chang Bao Han, Jianhua Huang, Yichuan Chen, Xiaobo Zhang, Xiaoqing Chen, Yongzhe Zhang, Hui Yan

**Affiliations:** 1Faculty of Materials and Manufacturing, Beijing University of Technology, Beijing 100124, China; yansx@emails.bjut.edu.cn (S.X.Y.); lezhi2005@163.com (Y.C.); ZhangXB@emails.bjut.edu.cn (X.Z.); chenxiaoqing@bjut.edu.cn (X.C.); hyan@bjut.edu.cn (H.Y.); 2College of Petrochemical Engineering, Hunan Petrochemical Vocational Technology College, Hunan 414012, China; huangjhyy@163.com

**Keywords:** perovskite, MAPbBr_3_, single crystal, phenethylammonium iodide (PEAI), surface-confined process

## Abstract

Organic–inorganic perovskite single crystals are promising in the field of optoelectronics due to their excellent optoelectronic properties. However, the ion transport of perovskite precursor is poor in confined spaces, which results in difficulty in the preparation of perovskite single-crystal films. Herein, MAPbBr_3_ films consisting of square grains were fabricated by the surface-confined process using the organic molecule PEAI (phenethylammonium iodide). Under the effect of oversaturation gradient, PEA^+^ is combined with the surface of perovskite grain from top to side, which constrains the lateral growth of grains and induces a downward growth of perovskite, leading to the formation of square grains. With the improvement of concentration PEAI, the perovskite film exhibits a decreased side length of grains (from 0.98 to 12.96 μm) and increased grain number and coverage, as well as crystallinity. The perovskite single crystalline grain films with PEAI showed double photoluminescence (PL) emission peaks due to the existence of iodine-rich perovskite. This work may provide a practical way to fabricate high-quality perovskite films for perovskite photoelectronic devices.

## 1. Introduction

Organic–inorganic perovskite materials have attracted attention in the field of optoelectronics due to their strong visible absorption, narrow full-width at half-maximum (FWHM) emissions, high mobility, and long carrier diffusion length [1,2,3]. Recently, remarkable achievements have been made in the research of optoelectronic devices based on perovskite materials, including high-efficiency perovskite solar cells (PSCs) [4], light-emitting diodes (LEDs) [5,6,7], and high responsiveness photodetectors [8,9,10]. However, it is worth noting that the perovskite materials in most optoelectronic devices, even the best PSC or LED, are still polycrystalline [11,12]. The efficiency or property of perovskite optoelectronic devices is limited by nonradiative recombination, which is caused by a large number of defects in polycrystalline films [11,12,13,14]. Reports have demonstrated that perovskite single crystals have significantly higher carrier mobility and longer carrier lifetime compared with polycrystalline, due to their extremely low defect state density and high crystallinity [15,16]. Moreover, perovskite single crystals own higher color purity and broader light-absorption [17]. With these improved properties, perovskite single crystals can increase the performance of optoelectronic devices. So far, perovskite single crystals have been applied in optoelectronic devices and have obtained good performance. In 2018, Ma et al. [9] reported a photodetector based on perovskite single-crystal film with ultra-high sensitivity and photoconductive gain. In 2020, Bakr et al. [18] reported a solar cell based on high-quality MAPbI_3_ single-crystal films with an efficiency of up to 21.9%. Therefore, perovskite single crystals have promising potential to further improve the performance in the field of optoelectronics [19,20,21].

Unfortunately, although high-quality perovskite single crystals can be prepared by many methods, such as the conventional cooling method [22], antisolvent vapor-assisted crystallization (AVC) [23], and inverse temperature crystallization (ITC) [16,24], these methods often have no restriction on crystal growth, resulting in the formation of bulk perovskite single crystals. For perovskite single-crystal films, the solution-based space-confined method was developed to prepare perovskite single-crystal films with a controllable thickness, but the lateral growth of perovskite crystals is limited by the poor transportation of precursor ions in the micrometer-sized gap [25,26,27,28]. The unfavorable thickness of bulk perovskite single crystals and the difficulties in the preparation of large-area films limit the development of perovskite single-crystal optoelectronic devices [24,26,27,29,30]. In recent years, researchers have prepared perovskite films composed of single crystals by introducing organic molecules to regulate the growth of perovskite during the spin coating [7,31,32,33,34]. Gao et al. [34] reported a high-quality MAPbBr_3_ film consisting of micron-scale single crystals by introducing benzophenone as a crystallization agent. In 2018, Lee et al. [32] reported a simple method to grow MAPbBr_3_ films covered with nano-sized single crystals by introducing phenylmethylamine (PMA) and fabricated LEDs with an external quantum efficiency (EQE) of 12%. Compared with polycrystalline films, the perovskite film composed of many high-quality crystalline grains not only retains the advantages of single crystals but also is more suitable for the structure of optoelectronic devices. Phenethylammonium (PEA), a commonly used additive for passivating defects and preparing two-dimensional perovskite [11,13,14,35,36,37,38], has a molecular structure similar to PMA. However, the introduction of PEA^+^ to prepare perovskite films composed of single crystals has not been reported. Moreover, the growth mechanism of perovskite films composed of crystalline grains is still unknown after the introduction of organic molecules.

Herein, we develop a surface-confined process to fabricate MAPbBr_3_ films consisting of square grains by introducing phenethylammonium iodide (PEAI). During the growth of perovskite grains, the growth direction transformed from lateral to downward due to the constraint of PEA^+^, resulting in the formation of square perovskite grains. The size of perovskite grain can be adjusted from microscale to nanoscale with the increase of PEAI. Moreover, the homogeneity, grain number, coverage, and crystallinity of the perovskite film were improved after the addition of PEAI. In addition, photoluminescence (PL) spectra show two emission peaks, which is ascribed to the MAPbBr_3-x_I_x_ epitaxy growth on MAPbBr_3_.

## 2. Materials and Methods

### 2.1. Materials

Phenethylammonium iodide (PEAI), methylammonium bromide (MABr), and lead bromide (PbBr_2_) were obtained from Xi’an Polymer Light Technology in China. N, N-dimethylformamide (DMF) was purchased from Sigma Aldrich. All chemicals were used as received without any further treatment.

### 2.2. Synthesis of Perovskite Films

Perovskite films were prepared on ITO substrates by using a one-step process. ITO substrates were sonicate washed with detergent, 2-propanol, acetone, and absolute ethyl alcohol for 15 min, respectively. Then, it was dried by nitrogen and placed in UV-O_3_ for hydrophilic treatment before spinning. To prepare the perovskite precursor, PEAI, MABr, and PbBr_2_ were dissolved in DMF solvent with the molar ratio of x:1:1. The x is the concentration of PEAI and the concentration of PbBr_2_ and MABr was 1 mol/L. The perovskite precursor was spin-coated onto the prepared substrates at 1500 rpm for 60 s and then annealed on a hotplate at 80 °C for 15 min. The spinning and annealing processes were all carried out in the glove box.

### 2.3. Characterization of Perovskite Films

The crystal structure of perovskite was characterized by XRD using Bruker D8 Advance X-ray diffractometer with Cu Kα radiation (λ = 1.5418 Å) as the X-ray source. Scanning electron microscopy (SEM) images of perovskite films were obtained using the JSM-6510 operated at 15 kV. The Steady-State PL Measurement was performed by an optical spectrometer (Maya 2000 Pro, Ocean Optics) equipped with a 400 nm laser at a pulse frequency of 1 kHz. The PL spectra of different positions on the grain were characterized by Confocal Raman spectroscopy (Witec alpha 300). Energy-dispersive spectroscopy (EDS) of perovskite single crystals was carried out using a field-emission scanning electron microscopy (SEM, Hitachi, SU8010) equipped with an EDS detector operated at 5 kV. The ultraviolet-visible (UV-vis) linear absorption spectra were obtained using a Persee TU-1810 spectrophotometer.

## 3. Results

### 3.1. Perovskite Film with and without PEAI

Perovskite films were deposited on the substrate using a one-step spin coating and the schematic illustration of the preparation process is shown in Figure 1a. The perovskite precursors composed of PEAI, MABr, and PbBr_2_ in DMF were spin-coated on cleaned and hydrophile treated substrates at 1500 rpm for 60 s. During the spin coating, the yellowish perovskite film was gradually formed. After annealing at 80 °C for 15 min, the color of the perovskite film deepened due to the evaporation of the remaining solvent and further crystallization of perovskite. When PEAI (Figure 1b) was added to the perovskite precursor, the morphology of grains in the perovskite film transformed from irregular shape to regular cubic, as shown in Figure 1c,d.

### 3.2. Morphology Characterization

To study the influence of PEAI on the morphology, the perovskite films with various concentrations of PEAI were characterized by SEM images, as shown in Figure 2a and Appendix A. The perovskite film without PEAI showed irregular grain shape with different grain sizes, indicating that the nucleation and growth of the perovskite is random and uncontrolled during the spin-coating process. The grains in the perovskite film prepared with PEAI showed a square shape and smooth surface. With the increase of PEAI content in perovskite precursor, the number and coverage of square grains gradually increased while the size gradually decreased (Appendix A). The above results indicate that a high concentration of PEAI can retard the growth of crystalline grains and enhance the number of nucleation. The smaller aspect ratio for higher PEAI concentration perovskite grains means a stronger restriction of PEAI on crystal growth (Appendix A). To further characterized the variation of grain size, we calculated the grain size distribution for different concentrations of PEAI (Figure 2b and Appendix A). With the increase of PEAI concentration, the average grain size decreased from 12.96 to 0.98 μm (Appendix A). Besides, the range of the grain size also gradually decreased (from 22.15 to 1.36 μm), indicating that the addition of PEAI was beneficial to improving the uniformity of grains and the homogeneity of perovskite films.

### 3.3. Structural Characterization

The crystal structure of perovskite was characterized by XRD spectra (Figure 3). Figure 3a presents the XRD spectrum of perovskite films without and with 0.15 mol/L PEAI. The perovskite film without PEAI displayed diffraction peaks at 15.27°, 29.47°, 33.09°, 36.44°, 45.24°, and 52.87°, corresponding to (001), (002), (210), (211), (003), and (310) crystal planes of cubic perovskite structure, as reported in other literature [24]. The dominance of (002) peak indicates the preferred orientation of perovskite growth is (001). For the perovskite film with 0.15 mol/L PEAI, just three diffraction peaks ((001), (002), and (003)) indicate that perovskite growth is highly oriented rather than random and uncontrollable under the impact of PEAI. With the increase of PEAI (≤0.15 mol/L) concentration (Figure 3b), the XRD peak intensity gradually increases, which is attributed to the addition of PEAI enhancing the orientation growth of perovskite. With continuing to increase the PEAI concentration, the intensity of XRD peaks began to decrease and the FWHM of XRD peaks increased rapidly at high concentration of PEAI (Figure 3d). This indicates that the crystallization of perovskites decreases and the quality of grains diminishes due to the excessive PEAI (Figure 3d). It is worth noting that even if the concentration of PEAI reaches 0.3 mol/L, the diffraction peak of two-dimensional perovskite still does not appear in XRD spectra. In addition, the (002) peak moved from 29.48° to 29.39° with increase of PEAI concentration from 0 to 0.3 mol/L (Figure 3c,d). The movement of the diffraction peak may be ascribed to the increase of lattice spacing caused by the doping of iodide [39].

### 3.4. Optical Characterization

To investigate the influence of PEAI on the optical properties of perovskite films, the steady-state PL spectra excited by 400 nm laser were characterized (in Figure 4). The PL spectra for perovskite film without PEAI showed an emission peak at ~550 nm (~2.25 eV), which corresponds to the optical band gap of 3D MAPbBr_3_ (Figure 4a). Interestingly, the PL spectra showed two emission peaks when the PEAI was added to perovskite films. The appearance of the new emission peak indicates that the formation of new perovskite materials is due to the addition of PEAI. Figure 4b and c are the first and second PL emission peaks in detail, respectively. Compared with pristine perovskite film, the first emission peak redshifts with the increase of PEAI concentration, which demonstrates that the content of iodine (I) in the perovskite gradually increased leading to a decreased bandgap (Figure 4b and Appendix A) [39]. The second PL emission peak is located at ~720 nm (~1.72 eV) and does not move with the increase of PEAI concentration (Figure 4c). Since PEA cations do not affect the optical properties of perovskite and many previous reports have confirmed that the pure iodide perovskite based on MA cations (MAPbI_3_) exhibits an emission peak at ~800 nm [40], we confirm that the perovskite corresponding to the second luminescence peak is MAPbBr_3-*x*_I*_x_*. Therefore, we define the MAPbBr_3-*x*_I*_x_* corresponding to the second emission peak () as “iodine-rich perovskite”. In addition, the photoluminescence of MAPbBr_3_ (first emission peak) is weakened with the formation of iodine-rich perovskite (Figure 4a), which may be due to the absorption of iodine-rich perovskite. The absorbance spectra of perovskite films with various PEAI concentrations are presented in Figure 4d. The increasing absorbance of perovskite films may be due to the increase of absorbed area caused by the large coverage of grains. The redshift of the absorption edge, corresponding to a decreased bandgap, can be ascribed to the increase of I^−^doping with the addition of PEAI (Figure 4e,f).

In order to acquire the structure and optical performance of single perovskite grains, we characterized the elemental distribution and PL spectra at different locations for the perovskite grains. The SEM image of single perovskite grain with 0.15 mol/L PEAI and the corresponding X-ray energy dispersion spectrum mapping images are shown in Figure 5a, b. The distribution of Br and Pb corresponds to the shape and position of perovskite grains in SEM (Figure 5a), and they are uniformly distributed in the perovskite grains. However, C and I were distributed uniformly in the whole SEM image, even on the substrate. Although C is always present in samples prepared using organic solutions, the presence of MA cations and PEA cations in the solution was also excessive for perovskites and the distribution of C on the substrate was not uniform. Thus, we speculate that residual C is distributed on the surface of grains and substrates. For the distribution of I, we believe that the uniform distribution of I throughout the whole region may be due to the excess I residue.

Besides, the steady-state PL spectra of perovskite grains at different positions were characterized by 532 nm laser excitation, as shown in Figure 5c. Only the emission peak (~735 nm) of iodine-rich perovskite exists at the grain edge, while the PL spectral for grain center not only shows the emission peak of iodine-rich perovskite but also presents the peak of 3D MAPbBr_3_ (~565 nm) (Figure 5d). The above results indicate that the iodine-rich perovskite epitaxy grows on the surface of MAPbBr_3_ crystals, so the perovskite grains may have a core-shell structure (MAPbBr_3-x_I_x_/MAPbBr_3_). The formation of this core-shell structure may be related to the different formation order of perovskite. It has been documented both theoretically and experimentally that the Pb–Br bond is shorter and stronger than the Pb–I bond [41,42,43], and the concentration of Br in the solution is much higher than I. Therefore, we speculate that the probability of forming Pb–Br in the solution is much higher than Pb–I, leading to [PbBr_6_]^4−^ (MAPbBr_3_) formation first in solution, that is, MAPbBr_3_ formed first. After the concentration of Br decreases, Pb-I and Pb-Br begin to appear together, so the formation of MAPbBr_3-x_I_x_ is later than MAPbBr_3_. The detailed formation process can be seen in Appendix A. It should be noted that this conclusion needs further confirmation.

### 3.5. Growth Mechanism

In order to reveal the formation process of perovskite films, we characterized the variation of the perovskite during the natural volatilization by microscopic images (Appendix A). For samples that are only spin-coated for 5 s, the solution spreads on the substrate to form a liquid film without any perovskite grains due to the centrifugal forces (Appendix A), indicating that the solution is still unsaturated and the square grains are not pre-formed in the precursor solution. Since the observation cannot be performed during the spin coating process, we observed the formation process of perovskite in stationary samples instead of the spin coating process. With the increase of the sample standing time, the solvent volatilized continuously, and the nucleation and growth of perovskite driven by the supersaturation gradually occurred in the solution as shown in Appendix A. About 30 s later, most of the solvent evaporated, perovskite grains were formed on the substrate (Appendix A), and the residual solvent was eliminated during the annealing process. The formation environment of the perovskite in the stationary sample was different from the environment produced by spin coating (Appendix A), which led to variation in the morphology of the perovskite films. Therefore, we can confirm that the crystallization process of perovskite occurs in the spin coating process after the liquid film is formed.

Based on the above experimental results, we propose a possible surface-confined process to explain the growth mechanism of single perovskite grain, as schematically shown in Figure 6. Generally, the nucleation and growth of perovskite grain are triggered by the supersaturation of the precursor solution, which is caused by the volatilization of the solution (solvent evaporation) during the spin-coating process [24,44]. The initial nucleation of perovskite occurs at the liquid–air interface due to the rapid evaporation of the solvent molecules from the surface of the perovskite solution. Meanwhile, perovskite formed at the liquid–air interface serves as a template for crystal growth in the subsequent process (Figure 6(a1)). The perovskite grows toward the substrate–liquid interface because the precursor is underneath, which promotes the oriented growth of the perovskite. Recently, many reports have confirmed this growth mechanism [45,46,47,48]. Pure MAPbBr_3_ grows laterally as well as downward due to the isotropic growth of crystals in solution, and the growth of grain in solution is dominated by lateral growth due to the region with large supersaturation being mainly on the side of the crystal (Figure 6b). However, the growth of pure MAPbBr_3_ will be affected when grains are in contact with the liquid surface. The portion of the grain above the liquid surface does not continue to grow due to the absence of the growth unit, while the portion below the liquid surface continues to grow laterally along the grain surface (Figure 6(a3,b3)). Finally, the pure MAPbBr_3_ grows into an irregular shape. In the early growth process, the presence of a template leads to the strong orientation of pure MAPbBr_3_, but the orientation of the film is gradually disordered due to the free combination between grains.

The perovskite with PEAI showed strong orientation under the further restriction of PEA cations. The growth direction of perovskite is surface-confined by PEA^+^: the growth direction changes from lateral growth to downward growth due to the restriction of PEA^+^ on the surface and side of perovskite (Figure 6c). Previous work has demonstrated that different raw materials have different solubility in DMF, and materials with low solubility prefer to precipitate earlier from the DMF [45,49,50]. Therefore, the PEA^+^ precipitates later than perovskite compositions (MABr and PbBr_2_) due to the higher solubility of PEAI in DMF.

As shown in Figure 6d1, PEA^+^ first appears in the high supersaturation region above perovskite crystals after the nucleation of perovskite at the liquid–air interface and it combines with the surface of perovskite due to the electrostatic attraction between the ammonium cation and [PbX_6_]^4−^ anion [11,51]. Then, PEA^+^ gradually combines with the sides of the grains under the action of the concentration gradient. The combination of PEA^+^ and grains will confine the growth of grains because the bulky PEA^+^ ligand inhibits the subsequent binding of halide lead octahedron and grains. Therefore, the downward growth of grains is enhanced and the combination between grains is prevented, resulting in further enhancement of the orientation of the perovskite films and the increase of grain numbers (Figure 6(d2,d3)). In addition, the suppression of lateral growth also avoids the influence of liquid surface on grain growth. Compared with the lateral growth of perovskite without PEAI (Figure 6b), the downward growth prefers to form single crystals with regular square shapes and smooth surfaces due to the lattice integrity of perovskite during the grain growth (Figure 6(d3,d4)). Thus a surface-confined process using the additive of organic molecules may provide an effective way to fabricate perovskite single-crystal films.

## 4. Conclusions

In summary, we successfully prepared perovskite films composed of square grains by employing PEAI. The surface-confined process was proposed to grow perovskite by the restriction effect of PEA^+^ on the surface and side of grains during the environment of concentration gradient. The size and coverage of perovskite grains can be controlled effectively by the concentration of PEAI. With the addition, the perovskite changed from irregularity to regular square shapes with a high growth direction of (001). The single perovskite grain shows two different PL emissions, ~565 nm and ~735 nm, which is from the side and center of square grain, respectively. The two emissions can be ascribed to the presence of iodine-rich perovskite and the core-shell structure formed with MAPbBr_3_. This surface-confined process using an organic molecule assistant may provide a feasible method for fabricating high-quality perovskite grains and film for optoelectronic devices.

## Figures and Tables

**Figure 1 nanomaterials-11-00473-f001:**
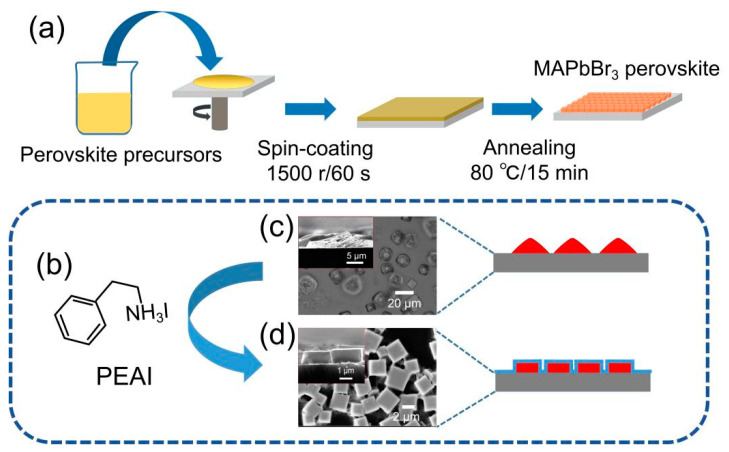
(**a**) Schematic illustration of perovskite film preparation. (**b**) The chemical structure of phenethylammonium iodide (PEAI). (**c**) SEM images of perovskite without PEAI. (**d**) SEM images of perovskite with PEAI (0.15 mol/L).

**Figure 2 nanomaterials-11-00473-f002:**
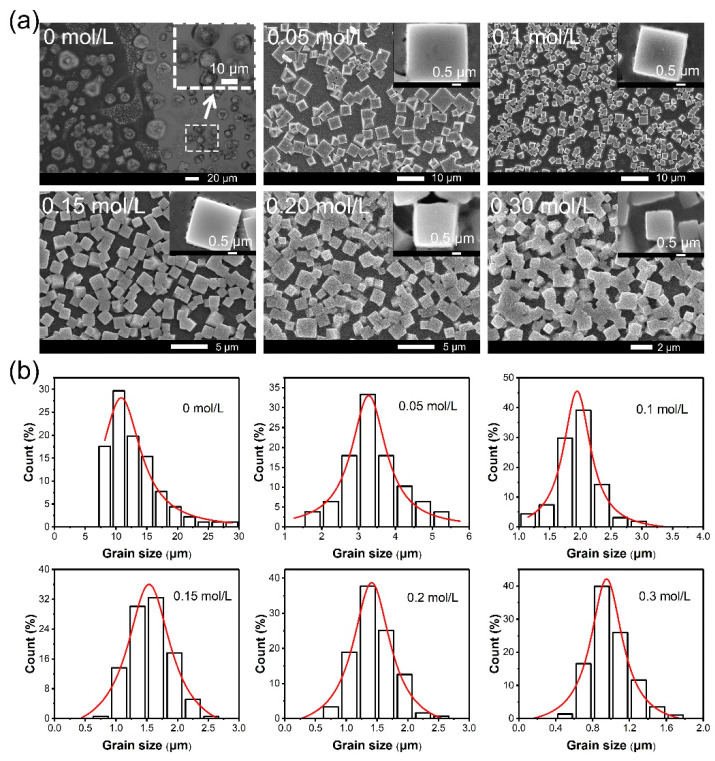
(**a**) Top-view microscopic images of perovskite films with various PEAI concentrations. (**b**) The grain size distribution of perovskite films at various PEAI concentrations.

**Figure 3 nanomaterials-11-00473-f003:**
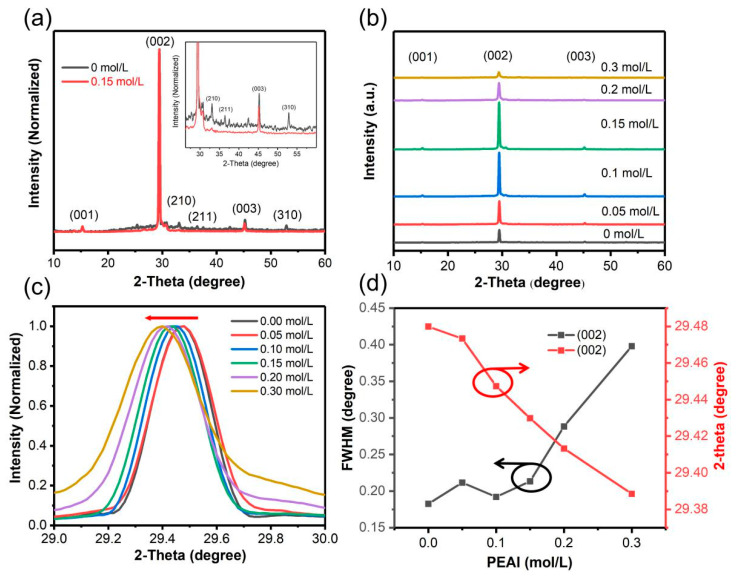
(**a**) XRD patterns of perovskite with and without PEAI. (**b**) XRD patterns of perovskite with various PEAI concentrations. (**c**) Normalized XRD spectra of (002) diffraction peaks according to PEAI concentration. (**d**) Full-width at half-maximum (FWHM) and intensity variation of (002) XRD diffraction peaks.

**Figure 4 nanomaterials-11-00473-f004:**
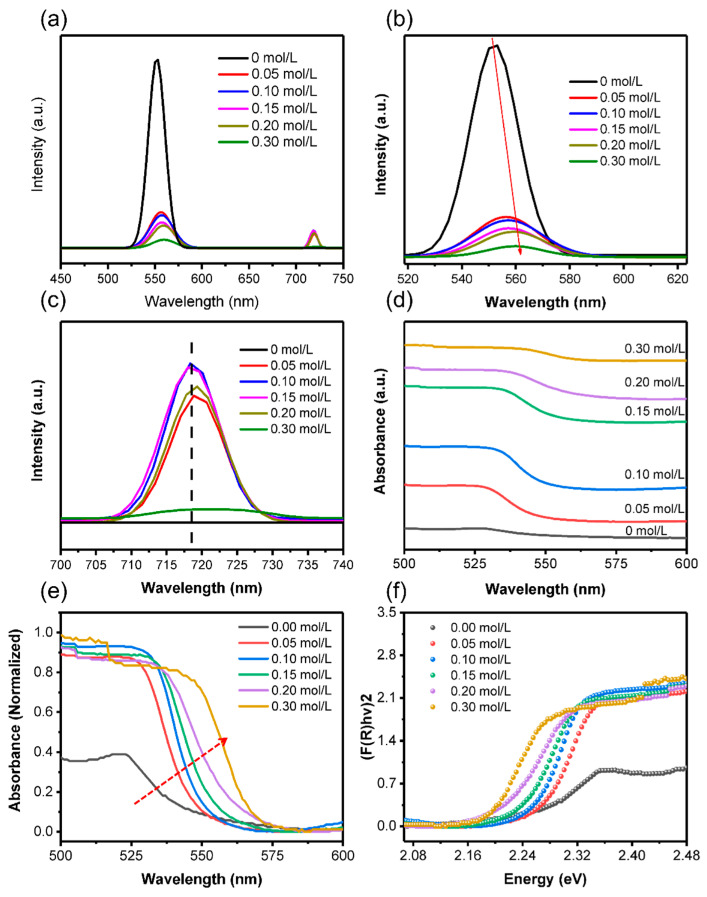
(**a**) PL spectra of the perovskite films with various PEAI concentrations. PL spectra of the first emission peak (**b**) and the second emission peak (**c**) at different PEAI concentrations. Absorbance spectra (**d**) and normalized absorbance spectral (**e**) of the perovskite films with various PEAI concentrations (**f**) bandgap energy of the perovskite films.

**Figure 5 nanomaterials-11-00473-f005:**
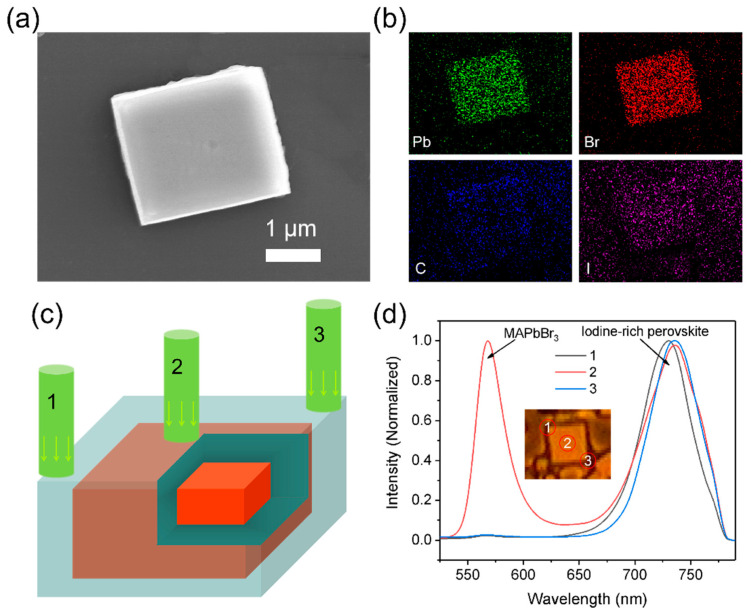
(**a**) Top-view SEM images of the perovskite with PEAI (0.15 mol/L). (**b**) Elemental mapping images corresponding to the SEM image. (**c**) Schematic diagram of perovskite grain excited by 532 nm laser. (**d**) PL spectral of perovskite grain with PEAI (0.15 mol/L) at different positions.

**Figure 6 nanomaterials-11-00473-f006:**
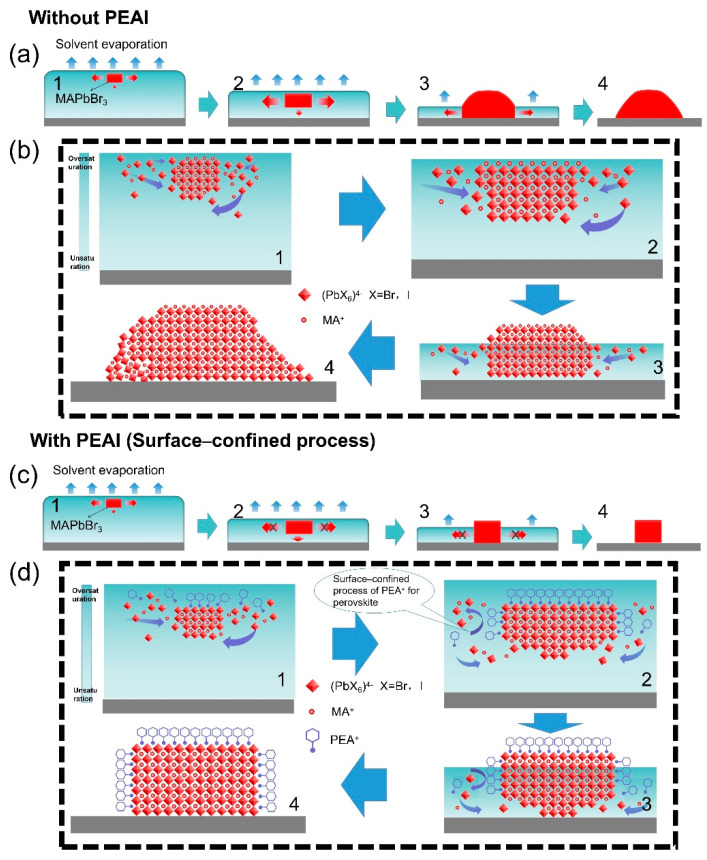
The schematic diagram of perovskite grains formation without (**a**) and with PEAI (**c**). (**b**) Detailed growth process of perovskite without PEAI. (**d**) Surface-confined process to growth perovskite grains. The four stages of perovskite crystallization are: (1) Nucleation of perovskite at the liquid–air interface; (2) The early stage of grain growth; (3) The later stage of grain growth; (4) The final formation of grain, respectively.

## Data Availability

The data that support the findings of this study are available from the corresponding author, upon reasonable request.

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
