# Peer review of "Organic Molecule Assisted Growth of Perovskite Films Consisting of Square Grains by Surface-Confined Process"

_nanomaterials, 2021, doi:10.3390/nano11020473_

Round 1

Reviewer 1 Report

In the paper by Yan et al., the authors propose a method to fabricate MAPbBr3 films consisting of cubic grains by adding phenethylammonium (PEA) iodide during the preparation process. In this way they achieve a lower growth rate that increases the nucleation centers, whereas the passivation of surfaces with the PEA cation imposes constraints in the lateral growth of the grains. As a result, highly oriented perovskite films containing grains with a regular cubic shape are formed. The paper is interesting and could be published in Nanomaterials upon revision with respect to the following comments:

1.      The addition of phenethylammonium usually gives rise to the formation of layered structures (e.g., see Energy Environ. Sci., 2018,11, 2188-2197). Can the authors exclude the presence of layered structures in their samples? Do the authors exclude that the second PL peak at 720nm is due to the formation of a 2D perovskite rather than an iodine-rich 3D perovskite phase?

2.      The authors argue that at high PEAI concentrations the XRD peak intensity decreases. Is this decrease of the XRD intensity due to the orientation disorder caused by the accumulation of grains, or maybe the quality of the grain material diminishes due to the incorporation of PEA and the subsequent formation of defects?

3.      Recent optical measurements based on spectroscopic ellipsometry have shown that the optical band gap of MAPbBr3 single crystals at room temperature is about 2.35 eV (J. Phys. Chem. Lett. 2020, 11, 7, 2490–2496), i,e, almost 0.1 eV higher than the PL measurement of the paper. How do the authors explain this difference?

Reviewer 2 Report

Shao Xin Yan et al. are reporting on growth control of MAPbBr3 by confining the surface using PEA cations. The growth method yields a preferential growth with control of the growing grain domains, from clear large ~cubic islands to almost conformal film. The results are investigated using different techniques, SEM/EDS, XRD, PL/absorbance. The experimental approach and results are interesting. However, the explanation of the process seems to need more improvement. Here come a few points that I would recommend to be considered:

Major:

  • The quality of the film without PEAI (pure MAPbBr3) seems bad coverage and already has preferential growth as shown in XRD. I would recommend a good explanation about bad coverage and pre-existing preferential growth.
  • Figure 3a, the signal to noise ratio is clearly different for with and without PEAI. It would be better to expand/enlarge Y-axis for both spectra with and without PEAI. As not only (001), (002), (003) peaks are there, as mentioned in line 153. If the spectrum for film with PEAI is expanded, several peaks will show up as well, however, indeed it seems to be reduced compared to without PEAI. So please use same eventual signal to noise ratios.
  • Line 164, the observed shift of the diffraction peak could be as mentioned related to iodine doping, however, looking the high concentration films, it seems also possible to be strained/stressed films upon usage of big PEA cations. Also, it could be a critical point for crystal structure change as more iodine is incorporated and then some tetragonal perovskites would be expected. Please provide explanation correlated with these possible options or why to exclude them.
  • Line 179–180, increased amount of iodine leading to decreased band gap…, is there any expected effect on band gap from PEA cations as well as that it is incorporated in the lattice? Please provide a clear explanation and reasoning to exclude this option or any possibilities for it.
  • Line 183, which pure iodine perovskite is discussed her? MAPbI3? No PEA incorporated? Please elaborate, specify, and cite relevant literature if possible.
  • Line 203, EDS, C and I distributed uniformly…, carbon is always observed every were in such solution prepared materials, and iodine diffusion from perovskites is known, it sounds like a weak correlation to conclude that the grains are covered by residual PEA MA and I.
  • The proposed mechanism for growth is not clear and not supported by experiments or literature, it needs a major improvement.

Minor:

  • Line 20, the use of the word “size” with linear values of “µm” to describe the size of “cubic grains”’ is not appropriate, it could be the length of the diagonal, thickness, or edge. In addition, the presented grains are not cubic. The crystal structure is cubic but the grains are rectangular. For published scientific results, I would highly recommend a proper choice of scientific words and correlated explanation. This applies to the whole manuscript.
  • Figure 1c,d , the SEM images are very small (difficult to see without enlarging the whole page), the presented area is also not similar, I would highly recommend the use of bigger images and similar presented areas for better comparison and understanding, same applies for the inset. Same applies to figure 2 with SEM images.
  • Line 114, “on pretreated substrates” what treatment?
  • Lines 160–161, not a clear sentence, please rephrase.

Round 2

Reviewer 1 Report

The authors have adequately replied to my comments and I can now recommend the paper for publication in Nanomaterials. 

Reviewer 2 Report

Thank you for the new version of the manuscript. I believe that the manuscript is better now and some confusion is cleared. Thank you for your efforts.